# Facial Emotion Recognition in Children and Adolescents with Specific Learning Disorder

**DOI:** 10.3390/brainsci10080473

**Published:** 2020-07-23

**Authors:** Francesca Felicia Operto, Grazia Maria Giovanna Pastorino, Maria Stellato, Lucia Morcaldi, Luigi Vetri, Marco Carotenuto, Andrea Viggiano, Giangennaro Coppola

**Affiliations:** 1Child Neuropsychiatry Unit, Department of Medicine, Surgery and Dentistry, University of Salerno, 84125 Salerno, Italy; graziapastorino@gmail.com (G.M.G.P.); stellatomaria92@gmail.com (M.S.); aviggiano@unisa.it (A.V.); gcoppola@unisa.it (G.C.); 2Department of Mental Health, Physical and Preventive Medicine, Clinic of Child and Adolescent Neuropsychiatry, University of Campania “Luigi Vanvitelli”, 80138 Naples, Italy; Marco.carotenuto@unicampania.it; 3Department of Clinical and Experimental Medicine, Child and Adolescent Neurology and Psychiatry, University of Catania, 95131 Catania, Italy; luciaa-@hotmail.it; 4Department of Health Promotion, Mother and Child Care, Internal Medicine and Medical Specialties (PROMISE), University of Palermo, 90133 Palermo, Italy; luigi.vetri@gmail.com

**Keywords:** facial emotion recognition, specific learning disorder, children, adolescents, executive functions

## Abstract

Background: Some recent studies suggest that children and adolescents with different neurodevelopmental disorders perform worse in emotions recognition through facial expressions (ER) compared with typically developing peers. This impairment is also described in children with Specific Learning Disorders (SLD), compromising their scholastic achievement, social functioning, and quality of life. The purpose of our study is to evaluate ER skills in children and adolescents with SLD compared to a control group without learning disorders, and correlate them with intelligence and executive functions. Materials and Methods: Our work is a cross-sectional observational study. Sixty-three children and adolescents aged between 8 and 16 years, diagnosed with SLD, and 32 sex/age-matched controls without learning disorders were recruited. All participants were administered standardized neuropsychological tests, evaluating facial emotion recognition (NEPSY-II), executive functions (EpiTrack Junior), and intelligence profile (WISC-IV). Results: Emotion recognition mean score was significantly lower in the SLD group than in the controls group on the Mann–Whitney U test for unpaired samples (*p* < 0.001). The SLD group performed significantly lower than the control group in their abilities to identify neutral expressions, happiness, sadness, anger, and fear compared to controls (*p* < 0.001). ER scores were positively correlated to the executive functions scores. There was no correlation with the Total Intelligence Quotient scores but there is a significant positive correlation with Working Memory Index and Processing Speed Index measured by WISC.IV. Conclusion: Our study showed that children and adolescents with Specific Learning Disorders have facial emotion recognition impairment when compared with a group of peers without learning disorders. ER abilities were independent of their global intelligence but potentially related to executive functions.

## 1. Introduction

In recent years, increasing attention has been given to the evaluation of social cognition (SC) and social functioning in children and adolescents, as this aspect is an important predictor of future school, work, and social successes and is crucial for adaptive functioning and a good quality of life [1,2].

Social cognition (SC) has been defined as the ability to understand, interpret, and respond appropriately to social cues in order to better interact with the external world, and it includes basic decoding abilities, such as facial emotion recognition (ER), as well as higher-order skills (Theory of Mind, empathy, and moral reasoning) [3].

Amongst all these abilities, facial emotion recognition is the ability to identify accurately the human emotions through the expression of face. Understanding facial expression is essential to correctly interpret the intents of others, modify self-behavior, and appropriately respond in social situations.

There are some innate emotions universally identifiable by humans, such as happiness, sadness, anger, fear, and disgust [4].

The ability to recognize these emotions, together with neutral facial expressions, develops gradually from childhood to adolescence: the first emotion identified is happiness, followed by negative emotions, such as fear, anger, and disgust [5,6].

In normal development, the neural networks for facial emotion recognition also mature progressively, from early childhood until the end of adolescence [7].

Neural networks underlying this ability involve a set of structures that includes the visual cortex, the orbitofrontal cortex, the insula, and the basal ganglia but mainly the mesial temporal structures, with an important role played by amygdala [8].

On the other hand, deficit in SC can significantly contribute to psychosocial difficulties in both children and adults [9,10].

Deficit in ER and in general SC have been typically highlighted in children and adults with Autism Spectrum Disorder (ASD), but also in other neurodevelopmental disorders such as Attention Deficit/Hyperactivity Disorder (ADHD), Specific Learning Disorders (SLD), Intellectual Disabilities (ID), or in many other different neurological and psychiatric conditions [11,12,13,14,15,16,17,18].

Specific Learning Disabilities constitute a diverse group of disorders in which children who possess normal intelligence have problems in processing information or generating output. These difficulties translate into a deficit in reading, writing, or calculating abilities (Dyslexia, Dysorthography and Dyscalculia) [19]; they can occur individually or more frequently together in the same subjects, with a prevalence of 2–5% in the school-age general population; the etiology of the disorder is multifactorial and reflects genetic influences or the dysfunction of brain systems [20,21].

Despite the limited number of research in this area, there are some studies showing that children and adolescents with SLD have a general impairment in recognizing emotions through facial expressions compared to their typical developing peers (TD) [22,23,24,25,26,27].

While all the different authors agree that SLD people are less accurate on tasks that assess a general understanding of facial expression emotion, there are is unequivocal consensus on how age, gender, and learning disorder subtype (verbal/non-verbal learning disorder) can affect these difficulties.

Furthermore, several authors suggest a potential association between ER skills and executive functions (EF); therefore, it is not clear whether they are distinct and functionally related abilities, or rather representative of a unitary process [28,29,30,31].

EF are high-level cognitive abilities supporting flexible behavior, adaptation to novel contexts, and an inhibition of stereotyped responses. They include basic skills such as attention, working memory, inhibitory control, and cognitive flexibility, and higher-order skills such as problem solving and planning [32,33,34].

Since a deficit in executive functions is common to many neurodevelopmental disorders and is a peculiar feature of SLD, it is possible that these functions affect the recognition performance of facial expressions in these subjects [35].

The objective of this study was to investigate ER skills in children and adolescents diagnosed with Specific Learning Disorders compared to an age/sex-matched group of peers with typical development.

Based on previous literature, it was hypothesized that the SLD group would perform worse in processing and correctly decoding facial expression stimuli compared with the control group. In addition, the secondary objective of our study was to assess the executive and cognitive functions in children and adolescents of the SLD group, in order to highlight a possible correlation with the ER abilities.

## 2. Materials and Methods

### 2.1. Sample Selection

Our work is a cross-sectional observational study that aims to explore facial emotion recognition in children and adolescents diagnosed with Specific Learning Disorder (SLD), and correlate them with cognitive and executive functions.

Sixty-three children and adolescents aged between 8 and 16 years with a diagnosis of SLD were prospectively recruited at the Child Neuropsychiatry Unit of the University Hospital of Salerno from December 2017 to December 2019.

The diagnosis was made by two independent child neuropsychiatrists with over 10 years’ experience in learning disorders, based on the clinical history and the standardized neuropsychological assessment of the intellectual profile and learning skills (reading, writing, and calculation tests), according to national guidelines for the diagnosis of SLDs.

A healthy control group, including 32 children and adolescents who attend the same hospital unit for a screening program of learning abilities, was also recruited.

In all patients of the control group, the diagnosis of SLD was excluded, and all had a normal cognitive profile. They performed the same standardized neuropsychological assessment of the SLD group.

SLD patients and controls were excluded if they had additional neurological (cerebral palsy, epilepsy, neurodegenerative diseases, or migraine), psychiatric (intellectual disability, attention deficit/hyperactivity disorder, specific learning disorder, autism spectrum disorder, anxiety, depression, and psychosis), or other relevant medical conditions (endocrinopathies, metabolic, hepatic, cardiac, or renal disorders).

To guarantee the homogeneity of the two groups, the variables age, sex, years of schooling, intellectual level, and level of maternal education were taken into consideration.

SLD patients and controls were administered a standardized battery tests by a single child neuropsychiatrist to evaluate cognitive abilities (Wechsler Intelligence Scale for Children fourth edition—WISC-IV), facial expression recognition (Nepsy-II), and executive functions (EpiTrack Junior).

All participants and their parents were provided a clear and detailed explanation about the purposes of the study and the procedures involved. Parents provided their informed consent in written form.

The procedure was approved by the local ethics committee, according to the rules of good clinical practice, in keeping with the Declaration of Helsinki. The procedure was approved by the Ethics Committee Campania Sud (protocol number N°179, 23/11/2018).

Sample characteristics are summarized in Table 1.

### 2.2. Facial Recognition Assessment—NEPSY-II

The Italian version of the NEPSY-II (Korkman, Kirk, and Kemp, 2011) is a battery of tests that aim to evaluate neuropsychological development in preschool, school age, and adolescent children [36].

It is composed of 33 tests that can be administered individually or with the entire battery. Our attention has been focused on the Emotions Recognition (ER) test that requires visually discriminating, recognizing, and contextualizing a series of facial emotional expressions.

The ER global results are expressed as raw scores; then, they are converted into age-weighted scores. The weighted scores are expressed by a numerical scale, with mean = 10 and standard deviation = 3. Scores below 7 are in the low range of the norm; scores below 4 were considered under the norm. For the analysis of individual emotions, we considered the error number.

### 2.3. Executive Functions Assessment—EpiTrack Junior

The EpiTrack Junior test is a standardized 12- to 15-min screening tool for executive functions assessment (Helmstaedter et al., 2010) in children and adolescents with epilepsy aged between 6 and 18 years [37].

It is administered individually, face to face, and consists of six subtests (Speed, Flexibility, Planning, Response Inhibition, Word Fluency, Working Memory) that contribute to determining a total score.

Standardization and correction for age resulted in a mean score of 33 ± 2 points, which was no longer correlated with age (r = 0.005). The retest practice effect was 0.7 ± 2 points, and the reliability r(tt) = 0.78.

A total score ≤ 31 points indicates an executive functions impairment, according to the following: 29–31 points = mild impairment; ≤28 points = significant impairment.

### 2.4. Cognitive Assessment—WISC-IV

The Italian standardization of the WISC-IV [38] was used. This tool is widely used to assess intellectual functioning in individuals from 6 to 16 years and 11 months old.

The WISC-IV consists of 10 core subtests, and 5 additional subtests that can be administered in addition to the core subtests to obtain more information on a child’s intellectual functioning, or as a substitute for the core subtests (see Manual).

The Italian version, according with the American standardization [39], allows us to calculate four main indices: Verbal Comprehension (core subtests: Similarities, Vocabulary, and Comprehension); Perceptual Reasoning (core subtests: Block Design, Picture Concepts, and Matrix Reasoning); Working Memory (core subtests: Digit Span and Letter-Number Sequencing); and Processing Speed (core subtests: Coding and Symbol Search). A Full-Scale IQ can also be calculated, which provides an overall measure of intellectual functioning.

The four indices and the Full-Scale IQ are expressed as age-weighted scores, with a mean = 100 and a standard deviation = 15.

### 2.5. Statistical Analysis

All neuropsychological scores were expressed as mean and standard deviation. The percentage of participants scoring lower than expected was also evaluated.

The normality of the distribution of analyzed parameters was assessed using skewness and kurtosis and the Shapiro–Wilk normality test.

In order to compare the demographic variables and the neuropsychological scores in the study group and in the control group, we used the Mann–Whitney U-test for independent sample or Chi-square test with Yates correction. A post hoc analysis of variance was performed by Kruskal–Wallis H-test.

The Spearman rank correlation test (two-tailed) was performed to evaluate the correlation between NEPSY-II, EpiTrack Junior, and WISC-IV scores.

The correlations were interpreted according to the guidelines adopted from Altman: r < 0.2, poor; 0.21–0.40, fair; 0.41–0.60, moderate; 0.61–0.80, good; 0.81–1.00, very good [40].

All data were analyzed using the Statistical Package for Social Science, version 23.0 (BM Corp. Released 2015. IBM SPSS Statistics for Windows, Version 23.0, IBM Corp: Armonk, NY, USA); *p*-values less than or equal to 0.01 were considered statistically significant.

## 3. Results

### 3.1. Sample Characteristics

Sixty-three children and adolescents (male = 34, 54%) with Specific Learning Disorders (mean age = 11.50 ± 2.82 years), and 32 age/sex matched controls were included in this study.

The two groups did not significantly differ in demographic characteristics (Table 1). The demographic and clinical characteristics of the participants (age, sex, and SLD types) are summarized in Table 1.

### 3.2. Executive Functions and Cognitive Profile

With regard to executive functions, 47/63 (75%) children with SLD obtained a score under the norm (<28), against 1/32 (0.03%) of the children of control group.

The EpiTrack Junior mean score of the SLD group fell into the “significant impairment” range (24.59 ± 4.53), while that of the control group fell in the normal range, showing significantly worse performance in the SLD group (*p* < 0.001; Table 2).

With regard the cognitive profile measured by WISC-IV, the mean Total IQ score fell within the norm for all the participants both in the SLD group and in the control group, with no statistically significant differences (*p* = 0.141, Table 2).

However, the WISC-IV sub-indices significantly differed between the two groups. In particular, the Verbal Comprehension Index and the Perceptual Reasoning Index were significantly lower in the control group (*p* = 0.008; *p* < 0.001, respectively), while the Working Memory Index and Processing Speed Index were significantly lower in the SLD group (*p* = 0.006; *p* = 0.009, respectively) (Table 2).

### 3.3. Emotion Recognition: Comparison between Groups and Correlation Analysis

Table 2 resumed all neuropsychological mean scores for the NEPSY-II, WISC-IV, and EpiTrack Junior in two groups, and the results of statistical comparison.

On the NEPSY-II ER (Emotional Recognition) subtest, 35/63 (56%) patients with SLD obtained a total score under the norm (age weighted score ≤ 4), against 0/32 (0%) controls.

The mean ER total scores also fell under the norm (<2 SD) for the SLD group (mean score 4.59 ± 2.52) while they were in the normal range for the control group (mean score 10.44 ± 1.00), and this difference was statistically significant to the Mann–Whitney U test for independent samples (*p* < 0.001; Table 2).

A post-hoc analysis did not reveal a significant difference in the mean ER scores between the three subtypes of SLD (*p* = 0.738).

Analyzing the number of errors in recognizing the individual emotions, the SLD group performed significantly lower than the control in their abilities to identify happiness, sadness, anger, fear, disgust, and neutral expressions compared to controls (*p* < 0.001 for all the emotion analyzed; Table 2).

The correlation analysis, carried out with the Spearman rank correlation test, showed significant positive correlations between ER scores and Epitrack Junior scores (r = 0.729, *p* < 0.001; Figure 1) in the SLD group.

The ER scores positively correlated also with the Working Memory Index (r = 0.569, *p* < 0.001) and the Processing Speed Index (r = 0.532, *p* < 0.001), while there was no significant correlation with the Verbal Comprehension Index (r = 0.009, *p* = 0.944) and the Perceptual Reasoning Index (r = −0.121, *p* = 0.347).

A statistically significant correlation between ER score and the Total IQ score has not been identified (r = 0.046; *p* = 0.718).

All the correlations between Nepsy-II (ER), EpiTrack Junior, and WISC-IV are summarized in Table 3.

## 4. Discussion

Our study investigated the emotion recognition through facial expressions in children and adolescents with Specific Learning Disorders, using a battery of standardized neuropsychological tests, and its the correlation with executive and cognitive functions.

In our study, children and adolescents with SLD showed an impairment in recognizing emotional states through facial expressions when compared to a group of typical development peers. These difficulties concerned the recognition of all the main human emotion (happiness, sadness, fear, anger, disgust) and were also associated with the incorrect tendency to attribute some emotional states to the neutral expressions.

In particular, in our sample, 56% of children with SLD obtained an ER score below the norm against 0% of controls, as measured by the NEPSY-II test. Furthermore, the statistical comparison between the mean scores of the two groups showed that the SLD group performed significantly lower than controls, both in global emotion recognition and in all the individual emotions analyzed.

An impairment of facial emotion recognition is described in many neurodevelopmental disorders, especially ASD, ADHD, ID, and in a wide range of child psychiatric conditions such as schizophrenia, mood disorders, anxiety, oppositional-defiant disorder, and conduct disorder [11,12,13,14,15,16].

Only a limited number of studies have been conducted in the pediatric population with SLD. Overall, our study is in agreement with the previous literature data, showing an impairment of the ER skills also in this population [23,25,26,27].

Holder et al. (1991), in a comparative study between children and adolescents with or without specific learning disabilities, showed that the group with SLD was less accurate in interpreting emotions and spent more time to perform the task than the control group. The authors also hypothesized that there were two SLD subgroups with different difficulties: the younger females with difficulty in interpreting the emotions and the older males who were rapid but often inaccurate [24].

In a more recent comparative study by Bloom and Heath (2010), the authors investigated the recognition and understanding of facial expressions of emotions in adolescents with general SLD and non-verbal SLD disorders compared to a control group [22].

The study showed that the group of subjects with general SLD was significantly less accurate in recognizing facial expressions than the group of subjects with non-verbal SLD and the control group, which did not differ significantly from each other.

In our study, the subtype of SLD does not seem to affect the emotion recognition ability; this could be due to the presence, in all our participants, of a verbal-learning deficit component, as well as to the low sample size. Further investigating this aspect may be the goal of our future research.

The social cognition skills in children diagnosed with SLD has been examined in a very recent study [41], in association with ASD and ADHD, through the analysis of Theory of Mind (ToM), which is another domain of social cognition that is defined as the cognitive capacity of an individual to be able to represent one’s mental states and others beliefs, desires, emotions, in order to explain and foresee the implementation of behaviors.

In this study, the authors showed that children with neurodevelopmental disorders such as ADHD, ASD, and SLD had ToM deficits independent of intelligence and language development, and there was a significant relationship between social cognition deficits and problems in social communication and interaction, attention, behavior, and learning.

Considering these results, it can be supposed that the difficulties in the emotion recognition through facial expressions is, in patients with SLD, partially responsible for the ToM deficit. In addition, also in this case, further studies need to investigate the correlation between both of these aspects.

In our study, the executive functions (focused attention, working memory, inhibition, cognitive flexibility, and verbal fluidity) are therefore compromised in SLD patients compared to controls, unlike the cognitive functions, which are normal. Indeed, 75% of the children in the SLD group showed a significant impairment in the EpiTrack Junior test. Moreover, the mean EpiTrack score of the SLD group was placed in the “significant impairment” range, while that of the control group was in the normal range, showing significantly worse performance in the SLD group compared to the control group (*p* < 0.001).

The analysis of the intellectual profile, carried out through the WISC-IV, showed that all the participants had a normal total IQ, and there were no significant differences between the two groups. However, significant differences emerged from the analysis of the individual WISC-IV sub-indices. In keeping with the previous literature, the SLD group presents greater difficulties in the Working Memory Index and Processing Speed Index, compared to the group of typical development peers.

Several studies support the evidence that WISC-IV intellectual functioning in children with SLD was characterized by a normal IQ with a significant discrepancy between general ability (Verbal Comprehension and Perceptual Reasoning Index) and cognitive proficiency (Working Memory Index and Processing Speed Index), which also occurs in our study [35,42].

In our study, the correlation analysis also showed that in the SLD group, ER abilities were significantly correlated to the executive functions skills assessed by EpiTrack Junior (r = 0.729; the strength of correlation was “good”). Therefore, our results highlight how patients with a greater impairment of executive functions are also those who encounter greater difficulties in recognizing facial expressions.

Furthermore, in the SLD group, ER abilities were significantly correlated also to the Working Memory and Processing Speed Indexes of WISC-IV, which represent a measure of executive functions, providing further confirmation of the relationship between executive functions and the recognition of emotions through facial expressions.

The correlation between executive functions and facial emotion recognition has been previously explored, and the results suggest a possible link between these two aspects in patients with different neurological and psychiatric conditions [43,44]. This correlation could have a double explanation: the impairment of executive functions could lead to a lower focused attention in performing tasks, as it has been demonstrated in Attention Hyperactivity Deficit Disorder (ADHD) patients; on the other hand, it could also be hypothesized that in patients with SLD, there is a common mechanism that affects the maturation and development of different neuronal circuits, which is indispensable for both executive functions and facial emotion recognition abilities.

More studies are needed to confirm these results. In particular, it would be interesting to integrate the neuropsychological evaluation with specific tests for selective attention to detail and visuo-perceptive abilities, such as the item “Completion of figures” of the WISC-IV battery. 

It could also be useful to analyze the key brain areas and structures activated both in the processes of facial emotion recognition and in the processes of writing, calculating, and reading compromised in SLD and in the executive functions, such as the prefrontal cortex and the cortico-limbic system.

The strength of the study is the use of a control group and standardized direct neuropsychological tests. This study has also many limitations. First, despite the use of standardized tests, facial emotion recognition is complex and difficult to quantify, and it is even more difficult to generalize to real-world contexts. Furthermore, our study does not take into account other aspects of social cognition such as Mind Theory (ToM), Social Decision-making, and Moral Cognition [19,20,45].

There are numerous future perspectives of investigation: emotion recognition could be evaluated in patients with different forms of SLD (verbal versus non-verbal) in a perspective way [46]; furthermore, imaging studies could be added in order to correlate neuropsychological alterations to functional and structural neuroanatomical data. The subsequent studies could also explore effects of emotion recognition deficit on children and their families’ quality of life.

## 5. Conclusions

Our results suggest that children and adolescents with Specific Learning Disorders have a deficit in facial emotion recognition abilities compared to their peers, independently of intelligence level. Our results also suggest that deficits in facial emotional recognition are potentially related to difficulties in executive functions, although it is not possible to establish a causal link between the two variables.

Since social skills such as the recognition of facial expressions are basic aspects for the correct development of social relationships, these aspects should be monitored in the developmental age, in order to guarantee children and adolescents with Specific Learning Disorders a good quality of life.

## Figures and Tables

**Figure 1 brainsci-10-00473-f001:**
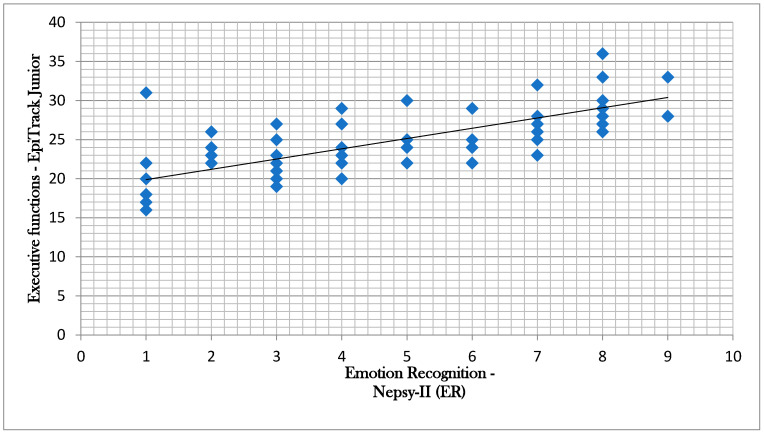
Correlation between Nepsy-II (Emotion Recognition) scores and EpiTrack Junior scores.

**Table 1 brainsci-10-00473-t001:** Demographic and clinical characteristics.

	SLD Group	Control Group	Statistics
**Sample size**	*n* = 63	*n* = 32	
**Sex**			
Male	34 (54%)	18 (56%)	
Female	29 (46%)	14 (44%)	Chi square test *p* = 0.833
**Age in years** (m ± SD)	11.50 ± 2.82	11.78 ± 3.87	U Mann-Witney *p* = 0.462
**SLD type**			
- Dyslexia, Dysorthography, and Dyscalculia	37 (59%)		
- Dyslexia and Dysorthography	17 (27%)		
- Dyslexia	9 (14%)		

SLD = Specific Learning Disorder; m = mean; SD = standard deviation.

**Table 2 brainsci-10-00473-t002:** Neuropsychological assessment in the SLD group and control group. SLD = Specific Learning Disorder; IQ = Intelligence Quotient; m = mean; SD = standard deviation; ** *p*-value < 0.01.

	SLD Group	Control Group	Statistics
	m ± SD	m ± SD	U Mann–Whitney
**NEPSY-II**	standardized scores		
**Emotion Recognition (ER)**	4.59 ± 2.52	10.44 ± 1.00	*p* < 0.001 **
	number of errors (raw scores)		
**Neutral**	1.96 ± 1.38	0.84 ± 0.88	*p* < 0.001 **
**Happiness**	0.72 ± 0.80	0.16 ± 0.37	*p* < 0.001 **
**Sadness**	3.72 ± 1.36	1.44 ± 1.80	*p* < 0.001 **
**Fear**	2.19 ± 1.38	0.50 ± 0.67	*p* < 0.001 **
**Anger**	2.70 ± 1.37	1.13 ± 1.1	*p* < 0.001 **
**Disgust**	2.36 ± 1.45	1.67 ± 1.10	*p* < 0.001 **
**EpiTrack Junior**	standardized scores		
**Total Score**	24.23 ± 4.34	32.22 ± 2.92	*p* < 0.001 **
**WISC-IV**	standardized scores		
**Total IQ**	95.62 ± 7.97	91.47 ± 10.61	*p* = 0.141
**Verbal Comprehension Index**	99.62 ± 11.98	92.88 ± 9.53	*p* = 0.008 **
**Perceptual Reasoning Reasoning Index**	102.87 ± 13.87	92.53 ±10.44	*p* < 0.001 **
**Working Memory Index**	87.62 ± 11.01	93.72 ± 10.00	*p* = 0.006 **
**Processing Speed Index**	87.59 ± 9.28	93.88 ± 11.04	*p* = 0.009 **

**Table 3 brainsci-10-00473-t003:** Correlation between Nepsy-II (Emotion Recognition), EpiTrack Junior and WISC-IV scores. TIQ = Total Intelligence Quotient; VCI = Verbal Comprehension Index; PRI = Perceptual Reasoning Index; WMI = Working Memory Index; PSI = Processing Speed Index; ** *p* value < 0.01.

	EpiTrack Junior	WISC-IV TIQ	WISC-IV VCI	WISC-IV PRI	WISC-IV WMI	WISC-IV PSI
NEPSY-II ER	0.729 **	0.048	0.009	−0.121	0.569 **	0.532 **
	0.000	0.709	0.944	0.347	0.000	0.000
EpiTrack Junior		−0.019	−0.095	0.045	0.495 **	0.490 **
	-	0.885	0.461	0.724	0.000	0.000
WISC-IV TIQ			0.523 **	0.397 **	0.151	0.136
		-	0.000	0.001	0.237	0.287
WISC-IV VCI				0.093	−0.120	−0.106
			-	0.468	0.350	0.406
WISC-IV PRI					−0.134	−0.221
				-	0.296	0.081
WISC-IV WMI						0.598 **
					-	0.000

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
