# Peer review of "Facial Emotion Recognition in Children and Adolescents with Specific Learning Disorder"

_brainsci, 2020, doi:10.3390/brainsci10080473_

Round 1

Reviewer 1 Report

The manuscript is entitled "Facial Emotion Recognition in Children and Adolescents with Specific Learning Disorder". The objective is to assess the emotional recognition abilities of children with learning disabilities and relate them to intelligence and executive functions.
Introduction
The introduction could be reorganized starting with the typical aspects (group the first paragraph together until "quality of life" then paragraphs 2 to 7) then the atypical aspects (start with the second part of the first paragraph "on the other hand, its deficits..." then paragraphs 8 and following).
When it says "Furthermore, several authors suggest potential association between ER skills and executive functions (EF)", references should be cited. The potential association between EF and ER skills should be better supported, probably by referring to work on autism.
The second objective of the study is to highlight correlations between EF and ER skills but not really to measure cognitive and executive functions.
Method
Why don't the authors propose a one-to-one match.
The choice of WISC-IV and not WISC-V should be justified.
Results
The results relating to executive functions and intelligence could be presented in the description of the population.
In this way, the results would focus on :
1- Emotion recognition: here, the results could be more precise by detailing the scores obtained by each typical LTC
2- Correlations between ER and EF: in the same way, more precise results could be given according to the type of EF (working memory, cognitive flexibility, inhibition, processing speed, verbal fluency, visual-spatial planning), as well as the WISC indices.
Discussion
The discussion could be resumed in the light of these additional analytical elements.

Line 104: sixty-three instead of "Sixty-tree".

Author Response

REVIEWER: 1

Reviewer comment: The introduction could be reorganized starting with the typical aspects (group the first paragraph together until "quality of life" then paragraphs 2 to 7) then the atypical aspects (start with the second part of the first paragraph "on the other hand, its deficits..." then paragraphs 8 and following).

Authors’response: We thank the reviewer for the suggestion. We reorganized the introduction as suggested by the reviewer, starting with the typical aspects and following with atypical aspects

Reviewer comment: When it says "Furthermore, several authors suggest potential association between ER skills and executive functions (EF)", references should be cited. The potential association between EF and ER skills should be better supported, probably by referring to work on autism.

Authors’response: We thank the reviewer for the comment. We added the following reference (28-31 in the manuscript):

  • Oerlemans A.M., Droste, K., van Steijn, D.J., de Sonneville, L.M., Buitelaar, J.K., Rommelse, N.N. Co-segregation of social cognition, executive function and local processing style in children with ASD, their siblings and normal controls. J Autism Dev Disord. 2013 Dec;43(12):2764-78. doi: 10.1007/s10803-013-1807-x.
  • Yang, C., Zhang, T., Li, Z., Heeramun-Aubeeluck, A., Liu, N., Huang, N., Zhang, J., He, L., Li, H., Tang, Y., Chen, F., Liu, F., Wang, J., Lu, Z. The relationship between facial emotion recognition and executive functions in first-episode patients with schizophrenia and their siblings. BMC Psychiatry. 2015 Oct 8;15:241. doi: 10.1186/s12888-015-0618-3.
  • David, D.P., Soeiro-de-Souza, M.G., Moreno, R.A., Bio, D.S. Facial emotion recognition and its correlation with executive functions in bipolar I patients and healthy controls. J Affect Disord. 2014 Jan;152-154:288-94. doi: 10.1016/j.jad.2013.09.027.
  • Förster, K., Jörgens, S., Air, T.M., Bürger, C., Enneking, V., Redlich, R., Zaremba, D., Grotegerd, D., Dohm, K., Meinert, S., Leehr, E.J., Böhnlein, J., Repple, J., Opel, N., Kavakbasi, E., Arolt, V., Zwitserlood, P., Dannlowski, U., Baune, B.T. The relationship between social cognition and executive function in Major Depressive Disorder in high-functioning adolescents and young adults. Psychiatry Res. 2018 May;263:139-146. doi: 10.1016/j.psychres.2018.02.046.

Reviewer comment: Why don't the authors propose a one-to-one match.

Authors’response: We thank the reviewer for the comment. We guaranteed the homogeneity of the two groups through the statistical comparison of age and gender (Table 1).

Reviewer comment : The choice of WISC-IV and not WISC-V should be justified.

Authors’response: We thank the reviewer for the comment. We used the WISC-IV because the WISC-V is not yet available in the Italian version.

Reviewer comment: The results relating to executive functions and intelligence could be presented in the description of the population. In this way, the results would focus on :

1- Emotion recognition: here, the results could be more precise by detailing the scores obtained by each typical LTC

2- Correlations between ER and EF: in the same way, more precise results could be given according to the type of EF (working memory, cognitive flexibility, inhibition, processing speed, verbal fluency, visual-spatial planning), as well as the WISC indices. 

Authors’response: We thank the reviewer for the suggestion. We reorganized Result section as suggested. We divided the Result section into three subsections: 3.1 Sample Characteristics; 3.2  Executive Functions and Cognitive Profile; 3.3 Emotion Recognition: comparison between groups and correlation analysis, adding the correlations with the sub-indices of the WISC-IV.

It was not possible to make the correlation with the individual executive functions because the  EpiTrack Junior does not provide the standardized scores for the individual functions but only for the total score.

We considered this aspect as a limitation of the study, and in future research we intend to use a more specific tool for the evaluation of the individual executive functions.

Reviewer comment: The discussion could be resumed in the light of these additional analytical elements.

Authors’response: We thank the reviewer for the suggestion. We modified the discussion in the light of the additional elements (correlations with the go sub-indices of the WISC-IV)

Reviewer comment: Line 104: sixty-three instead of "Sixty-tree".

Authors’response: We thank the reviewer for the comment. We apologize for the mistake which has now been corrected

Reviewer 2 Report

Overall, the study is clearly presented and the scope of the conclusions is within the results derived from the data. The manuscript would benefit from English language editing. There are some awkward phrasings and word usages that could be improved.

The authors list a number of exclusionary diagnoses. ADHD is not among these. Was ADHD considered in the evaluation for SLD? I am wondering if comorbid ADHD (common in SLD) is a potential factor affecting the NEPSY scores. ADHD might also contribute to the weakness on the EF tasks. If ADHD was not clearly assessed or ruled out in the SLD sample, this should be acknowledged more in the Discussion section and as a limitation. This would impact conclusions regarding the specificity of the finding and would need to be acknowledged.

I am not familiar with the EpiTrack Junior test, and I am presuming other readers might also not know this test. It would be good for authors to elaborate on it more. Is it individually administered via computer or face to face? Can the authors provide some reliability and validity information on the test? The scoring is also not clear – it appears scores are age-corrected, but not standardized? Further explanation would be helpful.

The authors report an a priori p-value for significance at .005 (line 177). This is highly restrictive and a bit unusual as 0.05 is more common. I am wondering if this is a typographical error.

In the Results section, the values in line 193 (mean and SD for NEPSY ER performance for SLD group) do not match the values in Table 2 for mean and SD for the ER group.

In the results section, the p-values in the text do not match the values in Table 2 (i.e., p<0.005 in text, p<0.001 in table).

The authors might consider reporting any post hoc analyses regarding SLD type. As the authors note, SLD is really a disparate group of disorders that may not share neural networks or substrates. Thus, lumping all types together does not provide as much information as individual types. The authors might consider a post hoc regression analysis to determine if there is one or another SLD type that has a significant role. Similarly, there are WISC factors that might also be considered post hoc. For example, is there potentially a relationship to one or another WISC factor such as WMI or PSI with outcome variables, even though FSIQ is not significantly correlated? This would be interesting to know, and clinically meaningful.

Table 2 would benefit from further clarification of score types for each of the variables/tests. For example, it would be good to clarify that the ER score is a scaled score, while the error scores for NEPSY are raw scores. This could be done in a legend attached to the Table.

The Discussion section is presented clearly and does not go beyond the data.  Limitations are presented appropriately and future research directions are also presented.

Author Response

(The authors gave the same response as above.)
